# Chasing Consistency: An Update of the *TCP* Gene Family of *Malus* × *Domestica*

**DOI:** 10.3390/genes13101696

**Published:** 2022-09-22

**Authors:** Mattia Tabarelli, Mickael Malnoy, Katrin Janik

**Affiliations:** 1Department of Agricultural, Food, Environmental and Animal Sciences, University of Udine, Via delle Scienze 206, 33100 Udine, Italy; 2Research and Innovation Centre, Fondazione Edmund Mach, Via E. Mach 1, 38098 San Michele all’Adige, Italy; 3Laimburg Research Centre, Institute of Agricultural Chemistry and Food Quality, Molecular Biology and Microbiology, Laimburg 6–Vadena/Pfatten, 39040 Ora/Auer, Italy

**Keywords:** *Malus* × *domestica*, *TCP* gene family, GDDH13v1.1 genome assembly

## Abstract

The 52 members of the Teosinte-Branched 1/Cycloidea/Proliferating Cell Factors (TCP) Transcription Factor gene family in *Malus* × *domestica* (*M.* × *domestica*) were identified in 2014 on the first genome assembly, which was released in 2010. In 2017, a higher quality genome assembly for apple was released and is now considered to be the reference genome. Moreover, as in several other species, the identified *TCP* genes were named based on the relative position of the genes on the chromosomes. The present work consists of an update of the *TCP* gene family based on the latest genome assembly of *M.* × *domestica*. Compared to the previous classification, the number of *TCP* genes decreased from 52 to 40 as a result of the addition of three sequences and the deduction of 15. An analysis of the intragenic identity led to the identification of 15 pairs of orthologs, shedding light on the forces that shaped the evolution of this gene family. Furthermore, a revised nomenclature system is proposed that is based both on the intragenic identity and the homology with *Arabidopsis thaliana* (*A. thaliana*) *TCPs* in an effort to set a common standard for the *TCP* classification that will facilitate any future interspecific analysis.

## 1. Introduction

The first complete genome of a plant, the model organism *Arabidopsis thaliana* (*A. thaliana*), was released at the very beginning of the sequencing era [1], and only ten years later, the milestone of the tenth plant genome being sequenced was achieved. The recent huge advances in genome sequencing technologies and the drastic decrease in sequencing costs have led to the generation of enormous amounts of primary and derived genomic data, with over a thousand plant genomes published in the past ten years (https://www.plabipd.de/index.ep, accessed 15 March 2022). 

The availability and ease of access to vast amounts of genomic sequences have enabled the proliferation of works aiming to identify gene families in a broad range of plant species, including members of the Teosinte-Branched 1/Cycloidea/Proliferating Cell Factors (TCP) gene family. *TCP*s consist of a group of genes found in all the land plants, from mosses to eudicots, that encode transcription factors acting as key regulators of a large number of biological processes related to several aspects of plant growth and development and, possibly, in biotic and abiotic stresses responses [2,3,4,5,6,7].

TCP proteins are characterized by the presence of a non-canonical basic helix–loop–helix (bHLH) motif known as the TCP domain, which mediates nuclear localization, DNA binding and protein–protein interactions [8]. Based on conserved differences in the TCP domain sequence that reflect functional differences, *TCP*s can be divided in Class I and II, the latter being further subdivided into Cincinnata (CIN) and Cycloidea/Teosinte Branched 1 (CYC/TB1) subclasses [3].

The evolution of the *TCP* gene family was shaped by several duplication events and a rapid expansion from lower to higher plants [2,9,10]. Consequently, a relatively small number of *TCP* genes have been found in the basal groups of land plants (e.g., seven in the moss *Physcomitrella patens* and six in the lycophyte *Selaginella moellendorffi* [11]), while higher plants exhibit the presence of a much larger number (e.g., 26 in *Oryza sativa*, 24 in *A. thaliana*, 46 in *Zea mays*) [9,12,13,14]. 

The identification of the *TCP* gene family in *Malus* × *domestica* (*M.* × *domestica*) dates to 2014, when Xu and collaborators [15] revealed 52 *TCPs* (*MdTCPs*) on the first apple genome assembly released in 2010 [16] (http://www.rosaceae.org; accessed 10 January 2022). The 52 genes were numbered from 1 to 52 based on the relative position on chromosomes and subdivided as follows: 22 in Class I, 26 in Class II-CIN and four in Class II–CYC/TB1. Additionally, the authors described 12 pairs of paralogs, of which seven were linked to potential chromosomal duplications associated with a genome-wide duplication, and two more pairs were tightly collocated in the apple genome. Therefore, it was proposed that the evolution of the *TCP* gene family in *M.* × *domestica* was shaped by both segmental duplication and transposition events [15]. 

In 2017, a new assembly of *M.* × *domestica* genome, GDDH13v1.1, was released [17] and is now considered to be the reference genome for apple. In fact, despite having the lowest number of predicted genes among all the previous *M.* × *domestica* assemblies, GDDH13v1.1 is characterized by an overall higher quality, making it the most complete among apple genome sequences published to date [18].

This study aimed to perform a novel identification of *MdTCP* genes based on the high-quality genome assembly GDDH13v1.1, which is followed by an analysis of the gene sequences to identify genes that originated from genome duplication events. Moreover, an *AtTCP*-homology-based nomenclature system for the *MdTCP* genes is proposed. 

## 2. Materials and Methods

### 2.1. Revision and De Novo Identification of the MdTCP Gene Family Set

Each of the 52 deduced amino acid MdTCP sequences identified by Xu et al. [15] on the 2010 assembly was used as a query for a tblastn search on the RefSeq database (http://www.ncbi.nlm.nih.gov/refseq/; accessed 10 January 2022) [19]. Additional *M.* × *domestica* TCP-containing putative protein sequences were obtained on Pfam (https://pfam.xfam.org/; accessed 10 January 2022) [20] and PlantTFDB (http://planttfdb.gao-lab.org/; accessed 10 January 2022) [21] databases. Alignments were performed on MEGA X software [22] (Mega X, version 10.2.5) with the Clustal Omega Multiple Alignment Tool [23] applying the following parameters: a gap-opening penalty of 5.00 and a gap extension penalty of 0.1 for the pairwise alignments; a gap-opening penalty of 10.00 and a gap extension penalty of 0.05 for multiple alignments; BLOSUM substitution weight matrix and a delayed divergent cutoff of 30% (sequences diverging by more than 30% were aligned later). All alignments were manually inspected and edited, if necessary. Selected sequences were subsequently used as queries to identify the gene positions on the GDDH13v1.1 assembly [17] (IRHS database; https://iris.angers.inra.fr/gddh13/; accessed 19 January 2022). The presence of an Open Reading Frame (ORF) and a TCP domain was determined on InterPro (http://www.ebi.ac.uk/Tools/pfa/iprscan/; accessed 25 January 2022) and Pfam databases. The GDDH13v1.1 Genome assembly and corresponding mRNA prediction list were accessed through https://www.rosaceae.org/analysis/242 (accessed 19 January 2022) and visualized using Blast and JBrowse functions. 

### 2.2. Identification of Potential Sister Genes among the MdTCP Gene Family

Reciprocal identities among MdTCP and AtTCP amino acid sequences were calculated on BioEdit software [24] on the alignment of amino acid sequences. The cutoffs for high, moderate and low-identity MdTCPs were set based on the Percentage of Identity (PID) values observed in AtTCPs.

To correlate the identity values with the pattern of chromosome duplication of apple, a Circos Plot [25] was generated on Galaxy servers (https://usegalaxy.eu/; accessed 7 February 2022) [26]. The Karyotype file of the GDDH13v1.1 assembly was downloaded from the IRHS database, while the BED file containing the relative chromosome positions of the putative paralogous *MdTCP*s was produced manually.

### 2.3. A Novel MdTCPs Nomenclature System Based on the Homology with AtTCPs

The *A. thaliana* homolog for each of the putative *MdTCP* was inferred based on DNA sequence similarity. A BLAST search restricted to *A. thaliana* sequences was performed using each deduced MdTCP amino acid sequence as query with the following cutoffs: query cover > 75% and Evalue < 1^ (−50). The same parameters were used to perform a Blast search restricted to *M.* × *domestica* sequences using AtTCP amino acid sequences as queries. 

The *MdTCP* neighbor-joining [27] phylogenetic tree was inferred on amino acid sequences with 1000 bootstrap reiterations. Evolutionary distances were computed using the Poisson correction method [28] in units of number of amino acid substitutions per site. The rate variation among sites was modeled with a γ distribution with a γ parameter of 1.00 and a homogeneous pattern, with the partial deletion option (i.e., fewer than 40% alignment gaps, missing data, and ambiguous bases were allowed at any position). Analyses were conducted with MEGA X software. 

To calculate the maximum likelihood tree on *A. thaliana* and *M.* × *domestica TCP*s, the amino acid sequences of 40 MdTCPs and 24 AtTCPs were aligned on MEGA X as described above, with subsequent manual revision. The Smart Model Selection algorithm [29] with the Akaike Information Criterion (AIC) was applied to calculate the appropriate evolution model. The maximum likelihood phylogenetic tree was inferred in PhyML 3.0 software [30] (Guindon; accessed from Bolzano/Bozen, Italy) using the Jones–Taylor–Thornton substitution model, discrete γ distribution with three parameters, proportion of invariable site, and empirical amino acid frequencies count (JTT + G + I + F). Initial trees for the heuristic search were obtained by automatically applying the Neighbor-Join and BioNJ algorithms to a matrix of pairwise distances estimated using the JTT model and finally selecting the topology with the superior log-likelihood value. 

## 3. Results

### 3.1. Revision and De Novo Identification of the MdTCP Gene Family Set

All the putative TCP sequences of *M*. × *domestica* found on public databases were aligned with the 52 *MdTCP* sequences published by Xu and colleagues [15]. The alignment showed that sequences retrieved from these databases correspond to the published *MdTCP* gene set, constitute fragments of *TCPs*, or they do not contain a TCP domain at all. The database search did not reveal the presence of other TCP-containing sequences besides those already mentioned in the *MdTCP* set [15], which were then employed for the subsequent analyses.

The manual evaluation of the 52 *MdTCP* sequences highlighted that four pairs (*MdTCP5*/*MdTCP6*, *MdTCP13*/*MdTCP14*, *MdTCP19*/*MdTCP20*, *MdTCP43*/*MdTCP44*) and one trio (*MdTCP8*/*MdTCP9*/*MdTCP10*) exhibit a reciprocal 100% identity. The accessions of these 11 sequences were aligned to the *M.* × *domestica* double haploid genome GDDH13v1.1 (IRHS database; https://iris.angers.inra.fr/gddh13/; accessed 19 January 2022) [17], and the members within each pair or trio mapped to single positions on the genome. Six of these redundant sequences, one for each pair and two for the trio, were thus excluded from the gene set. 

The alignment of the 46 remaining *MdTCP* sequences on the GDDH13v1.1 assembly resulted in 33 sequences matching with predicted protein-coding genes and 13 either mapping onto intergenic regions or partially overlapped non-TCP predicted gene regions. The corresponding GDDH13v1.1 nucleotide sequences of the 13 non-TCP mapped genes were therefore retrieved and verified for the presence of the TCP domain: seven deduced amino acid sequences (*MdTCP3*, *MdTCP4*, *MdTCP19*, *MdTCP36*, *MdTCP37*, *MdTCP41*, *MdTCP50*) display premature stop codons, and two more (*MdTCP7*, *MdTCP42*) constitute fragments of non-TCP gene sequences. The nine sequences were thus removed from the set. On the contrary, the remaining four non-TCP mapping sequences (*MdTCP2*, *MdTCP17*, *MdTCP23*, *MdTCP52*) contain a full-open reading frame (ORF) and a TCP domain and were thus kept in the set. 

Overall, 15 out of the 52 starting sequences were excluded from the set during the revision process: six due to redundancy, seven due to the presence of premature stop codons and two for constituting fragments of non-TCP genes. Table 1 summarizes the details regarding excluded and confirmed sequences, including previous and updated accessions. 

Thirty out of the 37 remaining sequences were found to be listed as TCP-containing in the mRNA gene list predicted from GDDH13v1.1. Additionally, three novel putative *TCP* genes that were not reported in the original set by Xu et al. [15] were identified in this predicted mRNA set. Three sequences (*MdTCP33*, *MdTCP40*, *MdTCP45*) previously observed to match with protein-coding genes on the GDDH13v1.1 assembly appeared to be automatically annotated as uncharacterized proteins. 

The three novel sequences include a start and a stop codon as well as a TCP domain. The sequences were provisionally named “nc” (“not classified”) 1, 2 and 3 (*MdTCPnc1*, *MdTCPnc2* and *MdTCPnc3*) and added to the set. The alignment of the deduced amino acid sequences of the three novel *MdTCPs*, displayed in Figure 1, shows that these three sequences are similar. In particular, MdTCPnc3 and MdTCPnc2 align, respectively, with the 3′ (positions 140–393) and 5′ end (1–265) of MdTCPnc1, which is the longest of the three. Consequently, the two shorter sequences MdTCPnc3 and MdTCPnc2 share an almost 100% conserved region of 124 amino acids, corresponding to the central region of MdTCPnc1. Interestingly, the two shorter sequences map on physically close regions of the same chromosome. 

Each TCP-mRNA predicted from the GDDH13v1.1 database was used as query for a BLAST search against the *A. thaliana* gene database “The *Arabidopsis* Information Resource” (TAIR; www.arabidopsis.org/aboutarabidopsis; accessed 22 February 2022) to retrieve all the GDDH13v1.1-predicted mRNA sequences displaying similarity to *AtTCP* genes. Out of a total of 24 *AtTCP* genes, only 14 were best hits for the 33 predicted *MdTCP* mRNAs. No additional mRNA sequence from the entire set of all mRNAs predicted from GDDH13v1.1 was found to be displaying similarity with any *AtTCP*, further confirming the absence of additional non-identified TCP-containing sequences in the GDDH13v1.1 genome. 

### 3.2. Identification of Potential Sister Genes among the MdTCP Gene Family

Potential sister *MdTCPs* were inferred based on the reciprocal identity values, namely the proportion of invariant sites for each pair of deduced amino acid sequences. Since recently duplicated *MdTCPs* are expected to show a higher percentage of identity (PID) than those that are non-recently duplicated, the *AtTCPs* PID was calculated to determine the threshold to define two genes as potential sister genes. *AtTCPs* exhibiting a notable PID have been identified based on the PID cutoff beyond which sequence alignment algorithms are able to distinguish between protein pairs of similar and non-similar structure (i.e., 35%) [31]. Among eight couples with a PID higher than 35%, the highest observed value was 62% and the mean was 47%. Percentages of 47% and 62% were thus selected as thresholds to define *MdTCP* sequences with moderate and high PID values, respectively. The identity matrix calculated on the deduced amino acid sequences of the 40 *MdTCPs* (Appendix A) is graphically represented in Figure 2. Results show that 14 pairs of *MdTCPs* share high PID (red squares), and one pair shows a moderate degree (light red). One trio (*MdTCPnc1*, *MdTCPnc2*, *MdTCPnc3*) exhibits a moderate PID: two sequences do not show a PID higher than the 47% threshold, but both show a PID higher than 47% with the third sequence, *MdTCPnc1*. Four sequences (*MdTCP2*, *MdTCP17*, *MdTCP23* and *MdTCP52*) appear to constitute a group with a moderate-high PID (mean 62%), making the relationships between each of the sequences in this cluster challenging to disentangle. Finally, three sequences do not display a moderate nor high PID with any other *MdTCP*. These results suggest that overall, at least 15 pairs of *MdTCPs* can be considered paralogous genes originated through the whole genome duplication event, although hints of a recent duplication appear in the three-member group, *MdTCPnc1*, *MdTCPnc2* and *MdTCPnc3*, as well as in the four-member group (*MdTCP2*, *MdTCP17*, *MdTCP23* and *MdTCP52*). 

The comparison between potential paralogous *MdTCP* genes and the apple intragenomic synteny is displayed as an overlapped Circos plot in Figure 3. Two *MdTCP* genes (*MdTCP2* and *MdTCP18*) map on the pseudochromosome 0, meaning that they were detected on contigs not assembled on any of the 17 apple chromosomes. Thus, the links involving these genes could not be represented. Figure 3 shows that the hypothesis of duplication of 15 pairs of genes is consistent with the results of the synteny analysis. Regarding the group of three, the two sequences *MdTCPnc2* and *MdTCPnc3* are located on the same chromosome (9), and both links with *MdTCPnc1* (indicated with red lines) are consistent with the duplication pattern (chromosomes 9–17). None of the links between members of the group of four genes (dotted lines) appear to be supported. Despite the similarity observed between the four sequences, the ambiguity of these results suggests that the four genes *MdTCP2*, *MdTCP17*, *MdTCP23* and *MdTCP52* cannot be considered as sister genes.

### 3.3. A Novel MdTCPs Nomenclature System Based on the Homology with AtTCPs

The *A. thaliana* homolog for each sequence was estimated through a BLAST query of each nucleotide sequence over the *A. thaliana* gene database to determine the best hits for each gene, which was followed by manual investigation and confirmation. *MdTCP* genes were thus re-named, appending the letter “a” after the number to differentiate between the current and the proposed nomenclatures. The letter “b” is used to indicate paralogs, which is in accordance with the previous analyses. In one case, three *MdTCP* sequences (*MdTCP12*, *MdTCP26*, and *MdTCP32*) showed similarity to the same *A. thaliana* sequence (*AtTCP9*), although only two of them exhibit a high percentage of reciprocal identity. To simultaneously highlight both the correspondence with the same *AtTCP* and the peculiar intragenic relationships, the two sister genes were named *MdTCP9a*/*b* and the third was named *MdTCP9-like*. Similarly, the cluster of four genes *MdTCP2*, *MdTCP17*, *MdTCP23*, and *MdTCP52*, whose evolutionary relationships are not exhaustively explained by whole-genome duplication origin, are similar to one *A. thaliana* gene only (*AtTCP17*): the four genes were therefore named *MdTCP17-like_a*/*b*/*c*/*d*. A unique *AtTCP* (*AtTCP1*) appears to be the best hit for the three genes *MdTCPnc1*, *MdTCPnc2*, *MdTCPnc3*, for which the genome duplication hypothesis partially explains the relationship between a gene and the other two. Thus, the three genes were named *MdTCP1a*/*b*/*c*.

In total, 19 *AtTCP* sequences were identified as homologs for the complete set of 40 *MdTCPs*. A *M.* × *domestica* ortholog was not found for the remaining five *AtTCPs*, namely *AtTCP11*, *AtTCP16*, *AtTCP22*, *AtTCP23* and *AtTCP24*. 

The complete revised set of *MdTCP* genes, listed in Table 2, comprises 40 unique sequences (full-length CDS in Appendix A), of which: 33 were deduced from both 2010 and GDDH13v1.1 assemblies.Four were deduced from the 2010 assembly but not annotated on GDDH13v1.1.Three were deduced from GDDH13v1.1 but absent on the 2010 assembly and, thus, not been predicted before.

The separation of *MdTCPs* into the two classes and the subclasses thereof is visible in the neighbor-joining phylogenetic tree displayed in Figure 4, panel A: 17 genes belong to TCP Class I (22 in the previous classification by Xu and co-authors [15]), 16 belong to the subclass CIN of Class II (26 in the previous classification) and seven belong to the Class II subclass CYC/TB1 (four in the previous classification). In *A. thaliana*, Class I contains 13 genes, Class II-CIN contains eight, and Class II-TB1/CYC contains three. 

In the maximum likelihood phylogenetic tree in panel B of Figure 4, including both *MdTCP* and *AtTCP* sequences, a general consistency between the *MdTCP*-*AtTCP* homologies previously determined can be observed. Even though the phylogenetic tree is unrooted, by implicitly rooting the tree at the split between Class I and Class II TCPs, it is possible to infer the relationships among genes, even though we advise the readers to refer to specific studies on phylogenetics and the evolution of *TCP* genes ([9,32]). While the Class I and the Class II-CIN subclass appear to be monophyletic, the phylogenetic relationships are not well defined in the subclass CYC/TB1. *AtTCP18* does not group with other members of Class II, and, in general, the node support of the CYC/TB1 taxon is low; thus, many nodes are represented as polytomies. This indicates that the intrinsic ambiguity in the alignment of the CYC/TB1 subclass sequences does not allow a confident reconstruction of the phylogenetic relationships in this group. 

## 4. Discussion

In the present work, the *MdTCP* gene family has been identified from the latest *M.* × *domestica* genome assembly GDDH13v1.1. Compared to the classification based on the previous draft genome sequence of 2010, the number of genes decreased from 52 to 40 as a result of the addition of three novel sequences and the exclusion of 15. The motivation for excluding these sequences was because nine did not exhibit a TCP-domain-containing ORF and six showed redundancy with other sequences. Interestingly, all the reciprocally identical sequences were characterized by sequential names, which reflect physical proximity on the chromosomes, since the previous nomenclature was based on the relative position of each gene on the chromosomes. Consequently, the duplications that led to the identification of redundant *MdTCP* sequences in the 2010 assembly are absent in GDDH13v1.1, and were likely to be artefacts generated during the in silico genome assembly. On the other hand, the analysis of the *MdTCP* percentage identity values and the comparison with the chromosome duplication pattern within the *M.* × *domestica* genome suggest that the origin of 15 pairs, and probably one more pair, is associated to chromosome duplication. As expected, the definition of the evolutionary relationship in the cluster of four genes is elusive and is further complicated by the fact that one member, *MdTCP2*, is mapped on the pseudochromosome 0. These findings suggest that the chromosomal or segmental duplication had a significant role in the generation of the *TCP* set in *M.* × *domestica*, while transposition events likely had a marginal impact.

Furthermore, in the present work, 19 *AtTCP*s were identified as homologs of the 40 *MdTCP*s genes. For comparison, in the classification carried out on the previous draft of the *M.* × *domestica* genome assembly, only 15 *AtTCP*s resulted as homologs for the complete set of *MdTCP* genes. Nonetheless, despite the higher number of *AtTCP* homolog genes found in the present work, no correspondent *MdTCP* was found for any of the five remaining *AtTCP*s. Interestingly, two paralogs have been found for each Class II *AtTCP* except *AtTCP24*, while several members of Class I appear to have only one or no ortholog in *M.* × *domestica*. In addition to highlighting the significant role of genome-wide duplication in the generation of the *MdTCP* gene family, these results suggest that further members of the *MdTCP* gene family, and especially Class I members, may have been subjected to a process of gene loss or have still to be identified.

The *TCP* nomenclature is usually based on a numbering system associated with the homology of the sequences with the corresponding *A. thaliana TCP*, but more often, *TCPs* (as well as other gene families) are named after the relative order of the sequences on the chromosomes. The adoption of two different standards to classify newly discovered *TCP* sequences is prone to ambiguities where identical TCP-containing sequences are indicated with two or even three different numbers. Such a condition constitutes a major obstacle for research: more and more genes are being characterized at a functional level in physiological pathways, interaction partners or expression patterns; however, the transferability of data among different species is severely hindered. An effort should be made to establish a common standard for the nomenclature of *TCPs*, also considering the exponential growth of data generation. In the present study, the identified *TCP* genes have been renamed after the homology with the *AtTCP* counterparts. In our view, such classification will facilitate future research by reducing ambiguities associated with a random numbering of *TCPs* in different species.

In the future, the availability of genomes of even higher assembly quality or pangenomes, transcriptome analyses and the integration of functional data will probably provide further details for a better understanding of the evolution and the function of the members of this important transcription factor family.

## Figures and Tables

**Figure 1 genes-13-01696-f001:**
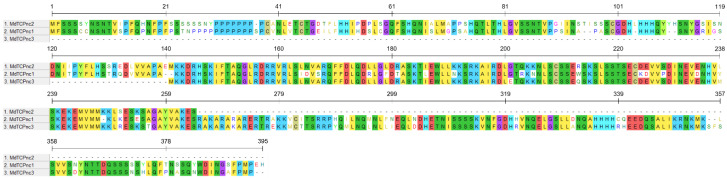
Alignment of the deduced amino acid sequences of the three novel putative *MdTCPs*, provisionally named *MdTCPnc1*/*2*/*3*. Numbers indicate the alignment positions, including gaps. The alignment was performed and visualized on MEGA X software. A colored background (MEGA X default color code) indicates identical positions in at least two sequences.

**Figure 2 genes-13-01696-f002:**
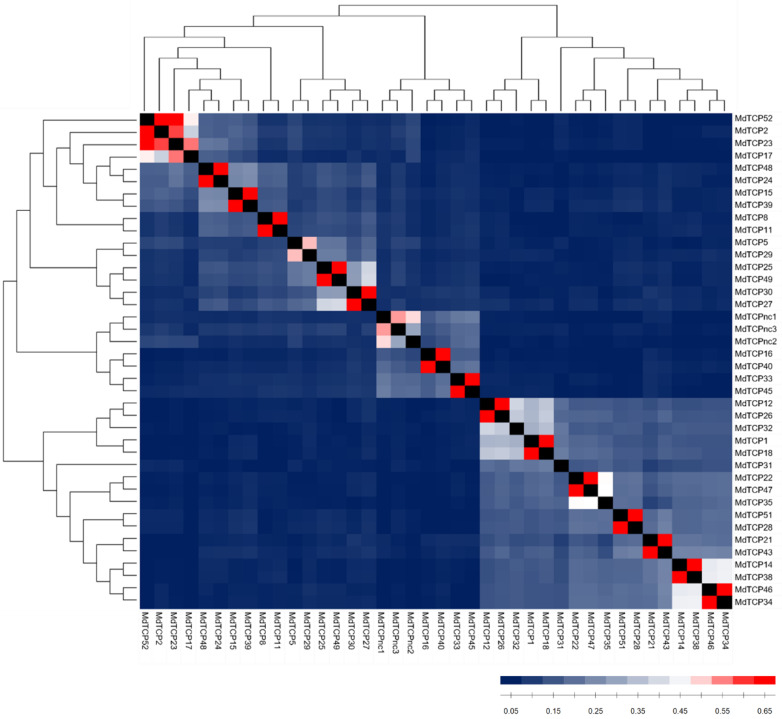
Graphical representation of the identity matrix calculated on the amino acid sequences of the 40 *MdTCPs* identified. MdTCP identifiers are reported to the right and bottom of each row/column. The phylogenetic relationships among genes are displayed by a neighbor-joining tree on top and to the left. The color gradient of each square indicates a low (blue), moderate (white), or high (red) percentage of identity between two MdTCPs.

**Figure 3 genes-13-01696-f003:**
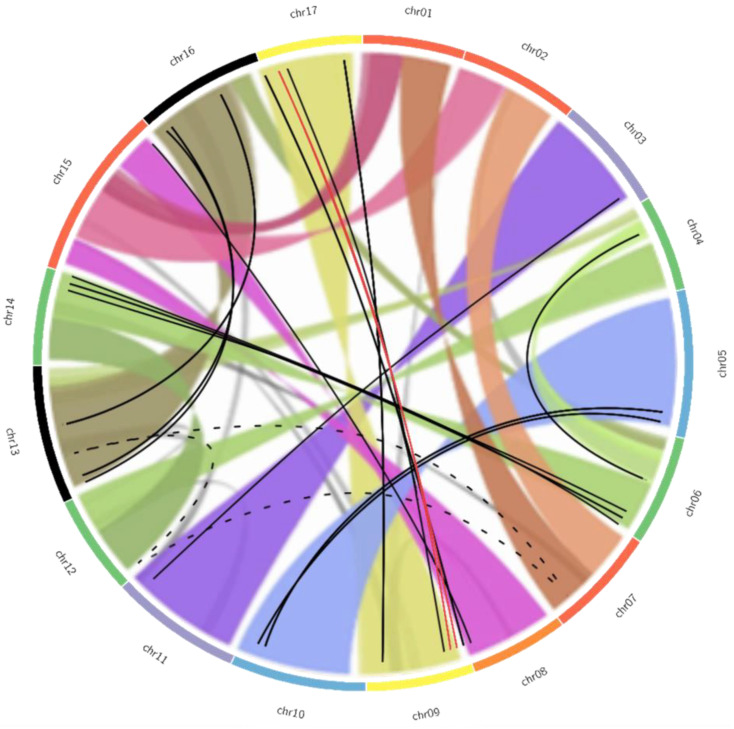
Graphical representation of the duplication pattern observed in *MdTCP* genes overlapped to pattern of chromosome duplication observed in the Apple genome. The pattern of apple genome duplication, as determined by Daccord and colleagues [17], is indicated as colored ribbons connecting homologous regions between the chromosomes, as represented by the colored slim boxes on the outside of the circle. Similar chromosomes are indicated with the same color to help visualization. Lines connect the physical positions on the chromosomes of the potentially duplicated *MdTCPs* in accordance with the previous analysis: the black solid lines link the couples, the red lines link the members of the group of three, while the black dotted lines connect members of the group of four. Adapted from Daccord et al., 2019 [17], under CC BY 4.0 license.

**Figure 4 genes-13-01696-f004:**
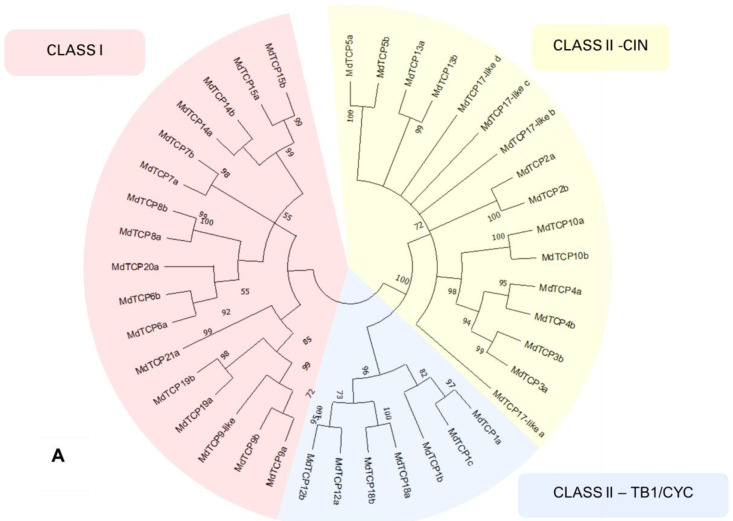
(**A**) Phylogenetic tree calculated on the 40 *MdTCP* genes classified in the present work. The different classes of *TCPs* are indicated with red (Class I), yellow (Class II-CIN) and blue (Class II–TB1/CYC). The tree was inferred using the neighbor-joining method [27] with 10,000 bootstrap re-iterations. Evolutionary distances were computed using the Poisson correction method [28] and are in the units of the number of amino acid substitutions per site. The rate variation among sites was modeled with a γ distribution. All positions with less than 60% site coverage were eliminated; i.e., fewer than 40% alignment gaps, missing data, and ambiguous bases were allowed at any position (partial deletion option). Node support is indicated as bootstrap values, and nodes with a support lower than 50 are collapsed. Analyses were conducted in MEGA X [22]. (**B**) Phylogenetic tree calculated from the amino acid sequences of the 40 *MdTCP* genes identified in the present work, and 24 *AtTCPs*. Members of the Class I are indicated with a red background, while those in Class II are indicated with a green background. Smart Model Selection algorithm [29] with Akaike Information Criterion (AIC) was applied to calculate the appropriate evolution model. The phylogenetic tree was inferred with the Jones–Taylor–Thornton substitution model, discrete γ distribution with three parameters, proportion of invariable site, and empirical amino acid frequencies count (JTT + G + I + F). Initial trees for the heuristic search were obtained by automatically applying the neighbor-join and BioNJ algorithms to a matrix of pairwise distances estimated using the JTT model and finally selecting the topology with the superior log-likelihood value (−37420,02). Nodes with a support lower than 50 are condensed. Analyses were conducted on PhyML 3.0 software [30] (Guindon; accessed from Bolzano/Bozen, Italy).

**Table 1 genes-13-01696-t001:** *MdTCP* genes identified by Xu et al. [15] and relative accession numbers determined on the 2010 genome assembly [16] and in the present work on the most recent genome assembly GDDH13v1.1 (IRHS database; https://iris.angers.inra.fr/gddh13/; accessed 19 January 2022) [17]. From this original set, six sequences were excluded due to redundancy with other sequences, seven due to the presence of premature stop codons and two for constituting fragments of non-TCP genes. A new GDDH13v1.1 accession was successfully determined for 33 *MdTCPs*, while for four additional sequences, an accession was not found.

Gene Name [15]	Accession Number Genome v1.0 [16]	Accession Number GDDH13v1.1 [17]	Reason for Exclusion
*MdTCP1*	MDP0000123919	MD01G1194100	
*MdTCP2*	MDP0000594000	Not found	
*MdTCP3*	MDP0000681033	Not found	Premature stop
*MdTCP4*	MDP0000393985	Not found	Premature stop
*MdTCP5*	MDP0000182310	MD05G1305100	
*MdTCP6*	MDP0000259723	nd	Identical to *MdTCP5*
*MdTCP7*	MDP0000534647	Not found	Fragment of non-TCP
*MdTCP8*	MDP0000927314	MD10G1259500	
*MdTCP9*	MDP0000920127	nd	Identical to *MdTCP8*
*MdTCP10*	MDP0000763497	nd	Identical to *MdTCP8*
*MdTCP11*	MDP0000287069	MD05G1281100	
*MdTCP12*	MDP0000280252	MD06G1070100	
*MdTCP13*	MDP0000877369	nd	Identical to *MdTCP14*
*MdTCP14*	MDP0000531313	MD06G1191800	
*MdTCP15*	MDP0000120671	MD06G1226300	
*MdTCP16*	MDP0000219838	MD06G1211100	
*MdTCP17*	MDP0000199422	Not found	
*MdTCP18*	MDP0000260056	MD00G1004900	
*MdTCP19*	MDP0000617746	Not found	Premature stop
*MdTCP20*	MDP0000253526	nd	Identical to *MdTCP19*
*MdTCP21*	MDP0000319266	MD08G1240000	
*MdTCP22*	MDP0000523096	MD09G1008300	
*MdTCP23*	MDP0000130524	Not found	
*MdTCP24*	MDP0000692406	MD09G1068200	
*MdTCP25*	MDP0000442611	MD09G1232700	
*MdTCP26*	MDP0000264920	MD04G1069300	
*MdTCP27*	MDP0000184743	MD03G1239100	
*MdTCP28*	MDP0000238683	MD13G1238400	
*MdTCP29*	MDP0000189749	MD10G1284400	
*MdTCP30*	MDP0000243495	MD11G1258900	
*MdTCP31*	MDP0000139807	MD12G1115100	
*MdTCP32*	MDP0000535805	MD02G1196100	
*MdTCP33*	MDP0000173048	MD13G1047200	
*MdTCP34*	MDP0000242185	MD13G1073500	
*MdTCP35*	MDP0000374900	MD13G1122900	
*MdTCP36*	MDP0000202241	Not found	Premature stop
*MdTCP37*	MDP0000693146	Not found	Premature stop
*MdTCP38*	MDP0000210785	MD14G1198200	
*MdTCP39*	MDP0000155433	MD14G1213400	
*MdTCP40*	MDP0000224810	MD14G1221800	
*MdTCP41*	MDP0000617459	Not found	Premature stop
*MdTCP42*	MDP0000247249	Not found	Fragment of non-TCP
*MdTCP43*	MDP0000515080	MD15G1431700	
*MdTCP44*	MDP0000608645	nd	Identical to *MdTCP43*
*MdTCP45*	MDP0000272980	MD16G1049000	
*MdTCP46*	MDP0000319941	MD16G1074800	
*MdTCP47*	MDP0000915616	MD17G1002500	
*MdTCP48*	MDP0000320363	MD17G1061100	
*MdTCP49*	MDP0000916623	MD17G1233500	
*MdTCP50*	MDP0000149841	Not found	Premature stop
*MdTCP51*	MDP0000851695	MD16G1243300	
*MdTCP52*	MDP0000373350	Not found	

**Table 2 genes-13-01696-t002:** Details of the 40 *MdTCP* sequences classified in the present work. The table lists the name of the genes determined on the homology with *AtTCPs* and the corresponding accession number and gene name as described by Xu and co-authors [15]. See Appendix A for full details.

Proposed Gene Name (*AtTCP*-Based)	GDDH13v1.1 Location ^1^	GDDH13v1.1 Accession	Current Gene Name [15]
*MdTCP1a*	Chr17:5918494..5919692	MD17G1073600	/
*MdTCP1b*	Chr09:5829036..5829824	MD09G1083300	/
*MdTCP1c*	Chr09:5866332..5867581	MD09G1083500	/
*MdTCP2a*	Chr10:35355926..35362503	MD10G1259500	*MdTCP8*
*MdTCP2b*	Chr05:41509298..41515345	MD05G1281100	*MdTCP11*
*MdTCP3a*	Chr03:32434147..32436258	MD03G1239100	*MdTCP27*
*MdTCP3b*	Chr11:37239267..37241971	MD11G1258900	*MdTCP30*
*MdTCP4a*	Chr09:29080345..29082126	MD09G1232700	*MdTCP25*
*MdTCP4b*	Chr17:28210166..28212381	MD17G1233500	*MdTCP49*
*MdTCP5a*	Chr06:35690088..35691110	MD06G1226300	*MdTCP15*
*MdTCP5b*	Chr14:29782365..29784375	MD14G1213400	*MdTCP39*
*MdTCP6a*	Chr09:580205..581231	MD09G1008300	*MdTCP22*
*MdTCP6b*	Chr17:174117..175642	MD17G1002500	*MdTCP47*
*MdTCP7a*	Chr13:24219020..24221263	MD13G1238400	*MdTCP28*
*MdTCP7b*	Chr16:26388679..26389521	MD16G1243300	*MdTCP51*
*MdTCP8a*	Chr08:30628021..30630739	MD08G1240000	*MdTCP21*
*MdTCP8b*	Chr15:53149356..53150897	MD15G1431700	*MdTCP43*
*MdTCP9a*	Chr06:16974738..16975295	MD06G1070100	*MdTCP12*
*MdTCP9b*	Chr04:9490696..9491838	MD04G1069300	*MdTCP26*
*MdTCP9-like_a*	Chr02:18841493..18842398	MD02G1196100	*MdTCP32*
*MdTCP10a*	Chr05:43753231..43753953	MD05G1305100	*MdTCP5*
*MdTCP10b*	Chr10:37444923..37446820	MD10G1284400 *	*MdTCP29*
*MdTCP12a*	Chr13:3317170..3318603	MD13G1047200	*MdTCP33*
*MdTCP12b*	Chr16:3451626..3453011	MD16G1049000	*MdTCP45*
*MdTCP13a*	Chr09:4657685..4660115	MD09G1068200	*MdTCP24*
*MdTCP13b*	Chr17:4956811..4958997	MD17G1061100	*MdTCP48*
*MdTCP14a*	Chr06:32752924..32754201	MD06G1191800	*MdTCP14*
*MdTCP14b*	Chr14:28826856..28828139	MD14G1198200	*MdTCP38*
*MdTCP15a*	Chr13:5176600..5177805	MD13G1073500	*MdTCP34*
*MdTCP15b*	Chr16:5227692..5228900	MD16G1074800	*MdTCP46*
*MdTCP17-like_a*	Chr13:13829138..13829581	/	*MdTCP23*
*MdTCP17-like_b*	Chr12:2031066..2031509	/	*MdTCP52*
*MdTCP17-like_c*	Chr00:20520105..20525532	/	*MdTCP2*
*MdTCP17-like_d*	Chr07:8098897..8101110	/	*MdTCP17*
*MdTCP18a*	Chr06:34380518..34383361	MD06G1211100	*MdTCP16*
*MdTCP18b*	Chr14:30342865..30345427	MD14G1221800	*MdTCP40*
*MdTCP19a*	Chr01:29290241..29291342	MD01G1194100	*MdTCP1*
*MdTCP19b*	Chr00:735426..737092	MD00G1004900	*MdTCP18*
*MdTCP20a*	Chr13:9108309..9109998	MD13G1122900	*MdTCP35*
*MdTCP21a*	Chr12:18364579..18365235	MD12G1115100	*MdTCP31*

^1^ In the event of sequences mapping onto a predicted gene, the location of the gene includes untranslated regions if present. * partially overlapping.

## Data Availability

All the sequences used and analyzed in the present work can be found on online repositories or databases: RefSeq (http://www.ncbi.nlm.nih.gov/refseq/; accessed 10 January 2022), Pfam (https://pfam.xfam.org/; accessed 10 January 2022), PlantTFDB (http://planttfdb.gao-lab.org/family.php?sp=Mdo&fam=TCP; accessed 10 January 2022), TAIR, (https://www.arabidopsis.org/browse/genefamily/TCP.jsp; accessed 10 February 2022), GDDH13v1.1 (https://iris.angers.inra.fr/gddh13/; accessed 10 February 2022).

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
