# Peer review of "Chasing Consistency: An Update of the TCP Gene Family of Malus × Domestica"

_genes, 2022, doi:10.3390/genes13101696_

Round 1

Reviewer 1 Report

The authors have made great efforts to provide an update of the TCP gene family of Malus x domestica by using some applications for sequence identification and alignment. The attained results may contribute to a better understanding of the TCP gene family of M x domestica and TCP gene family of other plant species. Generally, the current manuscript has been well documented and written. However, I have found some weak points which need to be revised and edited as below:

1. Materials and Methods should be specifically reworded and should be divided into subsections following the subsections narrated in the Result part

2. In line 94, page 2, the authors used AtTCP sequences for comparison with MdTCP sequences. However, in some published papers, some authors also used the TCP sequences in Arabidopsis, rice, chickpea etc., for generating the phylogenetic tree. It may be valuable to include a phylogenetic analysis of TCP transcription factors in M  x  domestica, Arabidopsis and some other species

3. In line 100, page 3, did the authors also check the phylogenetic relationship with other methods such as maximum likelihood?

4. Fig 1, Fig 3, and Fig 4 should explain the different colors performed by MEGAX software, etc.

5. Language editing still needs to recheck, for example; line 33-34, "has" should be "have"; line 221 page 7, "both link" should be "both links"

Reviewer 2 Report

Manuscript ID genes-1897812- entitled “Chasing consistency: an update of the TCP gene family of Malus × domestica” focused on the discoveries of novel Teosinte-Branched 1 / Cycloidea / Proliferating Cell Factors (TCP) using the available genomic sequences. For purpose, the authors used methodologies that are scientifically sound and an appropriate experimental design leading to the identification of 40 TCP instead of the 52 previously described. Based in these findings, the authors suggested a new nomenclature system refering to intragenic similarity and the homology with Arabidopsis thaliana TCPs. Despite this significant acheivement, therefore minor modifications are required.

Specific comments

Please see the attached PDF file.

Reviewer 3 Report

Tabarelli et al. in their research article entitled ‘Chasing consistency: an update of the TCP gene family of Malus × domestica’ conducted genome-level characterization and update of TCP genes in apple based on Malus x domestica GDDH13 v1.1 genome from a 'Golden Delicious' doubled-haploid tree (GDDH13), released in 2017. Although the TCP gene family has been reported previously, with the development of sequencing technology and the update of genomic data, some new genes or erroneous annotations will be discovered or corrected. The study is a knowledge updating case that may be helpful to researchers working in the field. However, I have some important concerns for this manuscript, the comments as follows:

1. There are several apple genomes in the GDR database (e.g. Malus x domestica Gala haploid v1.0 genome, Malus x domestica Gala diploid v1.0 genome, Malus x domestica HFTH1 Whole Genome v1.0, Malus x domestica GDDH13 Whole Genome v1.1, Malus x domestica Genome v3.0.a1, Malus x domestica Genome v2.0). Did the authors compare the number of TCP genes in these apple genomes? Whether their TCP changes are consistent with those in this article.

2. With the upgrade of sequencing technology, the genome version is updated, and the genome annotation information is more. Just as the work done by the author has updated the number of TCP gene annotations in Apple. The Arabidopsis genome is also being updated. Please authors explain that how to ensure the reliability and authenticity of the research results, the TCP genes were identified or named based on the Arabidopsis TCP genes.

3. In this manuscript, the authors have done some substantial fundamental work on the identification, classification, genome replication and naming of TCP genes. One suggestion is that some characteristic work on TCP gene family needs to be updated, such as tissue expression, stress response expression, and expression divergence of replication gene pair.

Round 2

Reviewer 3 Report

Many thanks to authors for responses. Although I have cautious about some responses, this study was no less than a timely update, and the main idea and results sections have been refined in revision. Overall, the data presented here would be interesting for the readers of this journal.